# Passing Networks and Tactical Action in Football: A Systematic Review

**DOI:** 10.3390/ijerph17186649

**Published:** 2020-09-11

**Authors:** Sergio Caicedo-Parada, Carlos Lago-Peñas, Enrique Ortega-Toro

**Affiliations:** 1Department of Physical Activity and Sport, Faculty of Sport Science, Regional Campus of International Excellence “Campus Mare Nostrum”, University of Murcia, 30107 Murcia, Spain; eortega@um.es; 2Faculty of Physical Culture, Sport and Recreation, Universidad Santo Tomás, Campus Piedecuesta, Santander 681027, Colombia; 3Faculty of Education and Sport Sciences, University of Vigo, 36310 Pontevedra, Spain; clagop@uvigo.es; 4Sports Performance Analysis Association, 30107 Murcia, Spain

**Keywords:** passing networks, tactical action, football

## Abstract

The aim of this study is to examine the most significant literature on network analyses and factors associated with tactical action in football. A systematic review was conducted on Web of Science, taking into account the PRISMA guidelines using the keyword “network”, associated with “football” or “soccer”. The search yielded 162 articles, 24 of which met the inclusion criteria. Significant results: (a) 50% of the studies ratify the importance of network structures, quantifying and comparing properties to determine the applicability of the results instead of analyzing them separately; (b) 12.5% analyze the process of offensive sequences and communication between teammates by means of goals scored; (c) the studies mainly identify a balance in the processes of passing networks; (d) the variables allowed for the interpretation of analyses of grouping metrics, centralization, density and heterogeneity in connections between players of the same team. Finally, a systematic analysis provides a functional understanding of knowledge that will help improve the performance of players and choose the most appropriate response within the circumstances of the game.

## 1. Introduction

Modern football is characterized by the study of performance indicators, a scientific methodology with a holistic approach that takes advantage of the technological and digital revolution to obtain a set of data on events that occur during a match [1]. These performance indicators have become essential tools for objectively codifying the particularities of the physical, technical, tactical and psychological behaviors constituting individual and collective performance, allowing for an understanding of the inherent characteristics of dynamic systems and relationship structures [2]. This methodology has evolved in recent decades, helping coaches optimize resources in order to achieve the proposed objectives, providing information that allows for the design of appropriate learning environments to enhance the performance of football players [1]. Among the performance indicators in professional football, study of passing networks has become more widespread [3].

Passing networks are dynamic systems, composed of articulated and interactive units that allow the identification, quantification and evolution of the game over time, taking into account stochastic forces combined with the complexity of analyses which are typical of the organization of a team [4]. The network structure combines tools derived from graph theory, statistical physics, nonlinear dynamics and Big Data [5], making Network Science one of the most active fields of applied physics and mathematics, due to its synergy of biological, social and technological systems [6], examining the role of the individual and differentiating the team from the sum of all its players in search of successful synchronization. As argued by [7], identifying coordination among players is crucial to obtain objective information that helps modify training methods according to specific needs.

In this sense, few studies have focused on analyzing advanced performance indicators for a broad understanding of collective tactical actions [8], or on cases where the multifactorial procedure in each of these actions is identified with emphasis on the methods of observation [9], or where the moment by moment of their behavior is visualized, captured, processed and analyzed [10]. The digital revolution has helped us obtain the precise location, in situ or by recording, of the players [11], thereby gathering information for the adequate distribution of the training content based upon objective criteria that provide explanations regarding patterns in movement, i.e., both their own and the opponent’s, and previous and subsequent, within the field of play [12]. Additionally, the study of passing networks allows us to analyze temporal evolution data with the objective of characterizing individual and collective needs, stimulating tactical components to obtain a successful resolution capacity in concrete situations within specific environments [13,14,15].

Although several investigations have provided analyses of passing networks in football, different authors have pointed out that it is necessary to scrutinize their usefulness, applicability and especially their methodological control [4,13,16,17,18].

The aim of this review was to examine research on network analyses and factors associated with tactical actions in football through a systematic review in order to identify the most researched issues in an attempt to describe the methodologies and developments in this field.

## 2. Materials and Methods

### 2.1. Search for Articles and Inclusion/Exclusion Criteria

A systematic review was conducted following the PRISMA guidelines (Preferred Reporting Items for Systematic Reviews and Meta-Analyses) [19].

For the search, critical reading and evaluation of the articles, the electronic database “Web of Science” was used. The search terms were: “network”, associated with the words “football” or “soccer”. Specifically, within the term “Topic”, the following command was entered: (TS = network) AND ((TS = soccer) OR (TS = football)).

The search included publications up to September 2019. Initially, we searched by determining the following inclusion criteria: (1) data relevant to the research object network analysis/tactical action; (2) conducted with male football players; (3) written in English. Studies were excluded if (1) they contained data from other sports; (2) they did not provide any relevant data on network analysis/tactical action; (3) they were conferences; (4) they were exclusively mathematical studies without applicability to sport. Twenty-five articles were included. Figure 1 shows the selection process of the papers.

Two authors independently selected the abstracts and full text of papers that met the inclusion criteria. In case of doubt or disagreement among the authors regarding the selection of the studies, a third, more experienced reviewer was included who made the final decision. The reliability of this record was calculated using Cohen’s Kappa coefficient, obtaining minimum values of 0.99.

### 2.2. Assessment of the Quality of Studies

In order to assess the scientific quality of the studies included in the systematic review, we used the checklist provided in [20]. Subsequently, to determine the quality of each study, we obtained the percentages that defined the methodological quality by ranking the results as described in [21]: (1) low quality ≤50%; (2) good quality between 51% and 75%; and (3) excellent methodological quality for scores >75%. The recording of the methodological quality of the studies was performed by two trained observers, obtaining minimum values by means of a Kappa coefficient of Cohen of 0.95.

### 2.3. Data Extraction

After the analysis of the methodological quality of the study, the following group of variables, related to the “Type of study”, were analyzed from the total number of papers recorded: (a) institution where the research was carried out; (b) country; (c) name of the journal; (d) competitive level of the sample; (e) observation instruments used in the data recording; (f) type of instrument; and (g) statistical analysis. (See Appendix A for specific data)

Subsequently, the following variables related to the “Objective of the study” were analyzed: (a) variables associated with the subject matter of the study (characteristics of the network: level of connectivity between players of the same team, grouping coefficient, shots and goals scored, starting and ending zones of the offensive sequences, shooting zones, tactical behavior (tactical principles), centrality metrics, contributions of the players taking into account the space-time positioning); (b) independent variable (ball possession, goals scored, goals conceded); (c) data collection system (means or instrument used for data extraction); (d) key findings (relevant research results); and (e) the quality score of each of the selected articles.

The data were recorded by two judges independently. Subsequently, the interjudge reliability was calculated for the evaluation of the studies by means of the Kappa Coefficient of Cohen, obtaining minimum values of 0.99.

## 3. Results

### 3.1. Search Results

The initial review identified 162 titles in the electronic database already mentioned. In this process, 75 references were manually removed due to the title. The remaining 87 articles were selected on the basis of title and abstract to establish their relevance; this allowed us to discard 12 more studies from the database. The remaining 75 articles were examined in detail; 51 documents that did not meet the inclusion criteria were discarded. In the last phase, 24 articles were selected for extraction and analysis.

### 3.2. Quality of Studies

Analysis of the methodological quality of the articles determined that the average study quality was 75.8%, with no study scoring below 50%. Of the sample, 33.3% had excellent methodological quality (one study with 93.75%, one 87.5%, and six 81.25%), while the remaining 66.7% was considered to be of good quality (eight studies with 68.75% and another eight with 75%) (See Figure 2).

### 3.3. Basic Characteristics of the Studies Included in the Review

From a general perspective (see Appendix A for specific data), the literature review process showed that with 12.5%, the Instituto Politécnico de Coimbra and the University of Coimbra of Portugal were the institutions associated with the most research, followed by the University of Lisbon of Portugal and the Federal University of Minas Gerais of Brazil, with a total of 8.3%. The country with the most research was Portugal with 41.6%, and the journals with the highest number of publications are International Journal of Performance Analysis in Sport, and Plos One, comprising 12.5% of the articles analyzed.

According to the characteristics of the observation methodology (see Figure 3), 91.7% of the studies were conducted under direct observation and 8.3% indirectly. Additionally, 91.7% of the studies did not mention the reliability or the number of observers, while 8.3% indicated both factors. In none of the studies was an observation instrument designed; however, 83.3% used observation instruments for match analysis, while 16.7% used a single set of raw data.

### 3.4. Summary of Individual Studies

The instruments and variables studied on the tactical action evaluated the general measures of passing using different software (8.3% Amisco Pro^®^ system; with 20.8% SocNetV; with 4.1% Longomatch Software (Fluendo SA, Barcelona, Spain); FUT-SAT (Federal University of Viçosa, Viçosa, Minas Gerais, Brazil; University of Porto, Porto, Portugal; Federal University of Minas Gerais, Belo Horizonte, Brazil), Software Performance Analysis Tool, MatchViewer from ProZone (Prozone Sports Ltd., Leeds, UK), Software NodeXL (Belmont, CA, USA), Software Social Network Visualizer (University of Patras, Patras, Greece); 20.8% used the unique data set of official web platforms and 16.6% recorded television images); 70.9% of the studies used multivariate statistical analysis; and 29.1% used univariate analysis. Additionally, 91.66% of the investigations applied a descriptive type methodology of direct observation and 8.33% a quasi-experimental one; 12.5% used sportsmen and women and children’s federated sports teams as the sample type, and 87.5% used high performance (see Appendix A).

Table 1 shows the treatment carried out for each selected article, detailing chronologically the following critical components: publication year, complete citation of the article, sample indicated in the research, independent variable, associated variables, data collection system and key findings.

According to the results, 25% of the articles were published in 2018, 20.8% each in 2016 and 2019, 12.5% in 2017, 8.3% in 2015, and 4.1% each in 2011, 2012 and 2014.

The main topics of study were: (a) grouping coefficient (100%); (b) shots and goals scored (8.3%); (c) start and end zones of offensive sequences (29.1%); (d) shooting zones (20.8%); (e) centrality metrics (33.3%); (f) tactical behavior (tactical principles) (8.3%); (g) ball possession (12.5%) and (h) player contributions taking into account spatial-temporal positioning (16.6%), in pursuit of better sporting performance.

The objectives of the studies were diverse; however, most of them coincided in analyzing the interactions between players, organizing the information of the teams on the field, which allows them to anticipate the competition and generate more options to win, sometimes creating tools in order to detect a change.

The independent variables mark a clear trend of the pass as the main characteristic of the analysis of networks, allowing the quantification of individual behaviors (contribution of the players), collective (grouping coefficients, centrality metrics, temporal evolution) and zones of the field of play (shooting zones, zones of beginning and end of the offensive sequences, shots and goals scored from the location in the field).

The associated variables show a predominance towards the analysis of interactions or connectivity between players (central players, passing networks) and the successful and unsuccessful possession of the ball.

The data collection system showed unique data sets as the main tool. The semi-automatic, multicamera or video monitoring system, recorded matches, TV broadcast logging and the tactical evaluation system, were also widely used.

In the studies analyzed, the following key findings were identified: (a) The greater the number of passes or network connections, the greater the probability of success and team performance; (b) The offensive midfield area is the region that predominates in the contribution of goals scored; (c) midfielders as the leading players (central players) in the gestation process play and intermediate successful plays; (d) the construction of the game using the side zones determines the highest clustering coefficient and, when well-connected, can optimize the team’s performance.

Finally, in relation to the results, there are coincidences in quantifying the intrinsic characteristics of the game, unraveling the sequences of passing and their relational structures from a new perspective that makes it possible to assess each team according to its adaptability and evolution throughout a match.

In relation to the competitive level of the sample, the results indicate that, in federated children’s sport (12.5% of the studies), instructed tasks and position during small conditioned games have a certain influence, allowing them to experiment with cooperative behavior to achieve the proposed objective. In line with the above, studies on high performance (87.5%) matches indicate that midfielders become central players, since they stand out in the construction of the game by recovering, resuming and organizing the gestation and attack processes.

## 4. Discussion

The aim of this article was to examine the available research on network analyses and the factors associated with tactical actions through a systematic review. The results show a direct relationship between network analysis and tactical actions, the latter describing the characteristics of the athlete’s interactions in a given context. The importance of the passing network is confirmed as being fundamental in characterizing the profile of the player and team from a more dynamic analysis [25]. The articles revealed the need to obtain an informative network that helps to average the general behavior of the team throughout the match, allowing us to determine the evolution and adaptation of the rival team [42]; therefore, a passing network needs to be made at the right time to the best located player and with the right trajectory [43].

In this regard, tactical positions allow for the calculation of weighted (micro level), weighted intermediation and weighted proximity of player connections [23,44] and team interactions (macro level) by applying centrality measures by means of graphs [18,25,30]. In addition, adjacency matrices are constructed from a set of nodes per team, which are captured through the distribution of observed passes in which the players represent the nodes in an area of a divided field, and the number of passes between two nodes represents the weight of the edge [33].

### 4.1. Performance Context

With regard to the samples used, the studies showed that they were homogeneous in relation to the need to investigate particular movements within the randomness of the game [17]. Likewise, they allow for the establishment of organized structures with the possibility of generating disorganization and anticipating the response of the opposing team [4]. In the analyzed works, specific performance characteristics were identified, such as (1) analyses of passes, a performance indicator that leads to the creation of a passing network as a reliable method to evaluate the interactions between players of the same team [13,45]; (2) 37.5% of the investigations showed the need to analyze in detail the influence of performance [32] as a notion comprising key members [35], interaction between players, the type and strength of the connection between them and the contribution of each player according to his position and zone of the field, examining the role of each to understand the collective tactical performance [46]; and (3) interpreting the pass as a successful or unsuccessful connection to improve the construction of the tactical actions through the identification and understanding of individual patterns [41].

Orchestration in terms of game production suggests that a performance indicator is a successful mobilization of complex tasks constituting a pattern between player interactions, taking advantage of each other’s knowledge and capabilities [23]. Within the limitations of specific tasks, there is a lack of instructions by coaches and assigned tactical tasks as a result of some players creating outstanding bonds with some colleagues; thus, establishing behaviors for strategic planning is crucial [18].

### 4.2. Network Interaction and Variability

In this way, network analyses allow us to identify the location of players with values of density, total links and grouping coefficients between the connections made [34], the influence of ball possession [30,44], determining the center of mass of the team as a key player and the positional average of all players [26], establishing movement predictability, type of organization and dynamic assemblies based on the needs of the team’s tactical behavior, involving the shooting zones in relation to contextual variables within the offensive sequence to score a goal and explore goal opportunities [27,33,47].

Due to the inherent nonlinearities in team performance, the importance of understanding tactical versatility in different matches is emphasized [38,48]. It was found that 50% is a widespread opinion of research that ratifies the importance of the network structure, quantifying and comparing its properties in order to determine applicable results rather than analyzing them separately, thus complementing the vision of the game and avoiding the loss of information [10,22]. These information networks are a suitable model for the detection and analysis of each team’s playing patterns [7], i.e., performance indicators that explain the differences between winning, drawing or losing a match [37]. Among the studied research, 12.5% analyzed the process of offensive sequences and communication between teammates according to goals scored [49], in order to understand and improve behavior in general [16,18]. The great variability of player actions and interactions reveals how interpersonal interactions may change from one game to another, determining a prolonged relationship between passes that, if well connected, can optimize performance [13,18,22].

### 4.3. Specificity of Collective Tactics

In relation to the objectives and results, the studies analyzed mainly noted an equilibrium or improvement in the processes of the passing network, where the variables studied have made it possible to facilitate and interpret the analysis of grouping metrics, centralization, density and heterogeneity in the connections between players of the same team [31,34], determining factors to develop strategies and create new options for offensive tactical performance from the perspective of nonlinearity, generating an organizational structure with situations of superiority that allows them to anticipate and create surprise in the opponent.

After the description of the variables of the passing networks, the importance was emphasized of identifying the associations between codes that represent individual and collective patterns of play in the two networks competing for a common goal [40]. Thus, networks become the raw material of quantitative analyses for the computerized coding of the relational system; this allows for a more precise method to understand the functioning of the team [50], providing important information for the temporal predictability of the game. The importance of network characteristics in the adaptability of coaches to change team strategy and behavior according to the state of the game was demonstrated [33,51,52].

In the face of tactical applications, it is important to pose tasks in real game situations, e.g., building offensive maneuvers from lateral areas aimed at the recovery and possession of the ball, as well as goal scoring, through small and conditioned games that increase cooperative behavior and the focus of the network, which will help identify the importance of midfielders and the individual role of each player and their performance in the areas of the field.

### 4.4. Collective Team Behaviors

Network analyses provide coaches with meaningful information about the behavior of sports teams. Decision-making in training processes with specific individual and collective properties and conditions increases the level of synergy between players and team synchronization [53]. This allows for the readjustment of behaviors on the field while attacking and defending [54].

From a broader perspective, midfielders constitute the offensive phase with high values of density, degrees of centrality and total commitment, using short passes and covering all sectors of the field [28]. Network analysis provides significant elements for determining the key contributions of each player, identifying creative players and determining collective performance in order to organize team strategies [25,34].

Favorable connections between players for the execution and reception passes will increase the number of actions, and with it, the probability of success, allowing for the identification of the best strategy for the situational variables and the evaluation analysis to predict the future of the game, using key performance variables such as ball possession, interaction with central players and effective shot attempts.

On the other hand, the need arises to consider, for future lines of research, the individual and collective adaptation in the field of play in relation to the location of the ball, which will provide relevant and specific information on the interactions of the players within the dynamics of the game, in order to compare the performance with other studies and check the effectiveness in the development and implementation of possible effective strategies in the overall performance of the group.

### 4.5. Limitations

A possible limitation of the present study is evidenced by taking into account only research published in English and limiting the search to the Web of Science; although this database is outstanding at an international level in the dissemination of scientific knowledge, it does not cover all the publications made on our object of study. In consideration of the results of the research evaluated in this review, the need to carry out subsequent reviews in other languages and databases is recognized. Likewise, the results allow us to recommend linking conceptual and operational definitions of the variables under study in future research.

As for the studies examined, it is suggested that future research analyze the passing networks of the same team throughout a season, in order to achieve a deeper knowledge of the selected team and a clearer view of the variations of the networks depending on the rival. In the same sense, it is important that the results of the research provide practical information that allows coaches to implement strategies that improve performance, based on specific analyses of the position and function of each player according to their evolution in terms of performance.

## 5. Conclusions

It is important to note that network analyses are a novel tool that may be applied to explore the behavior of football players, identifying the key players in a match and measuring centrality and the probability of passing success in different areas of the field.

The results reveal that performance can be optimized according to the position of the player, experiencing different roles that allow him to adapt in terms of cooperation and balance, in the formation of synergies and couplings that support the synchronized movements of the team. In this context, high values in the connectivity metrics demonstrate the usefulness of identifying the relationships between players of the same team and determining the causes of failed passes, generating new tactical skills and systematic creativity within the individual and collective resolution capacity. Finally, this information should focus on the contextualization of data to design training tasks with different game formats. Finally, systematic analyses provide knowledge that will help improve the performance of players and inform choices of the most appropriate response within the circumstances of the game.

## Figures and Tables

**Figure 1 ijerph-17-06649-f001:**
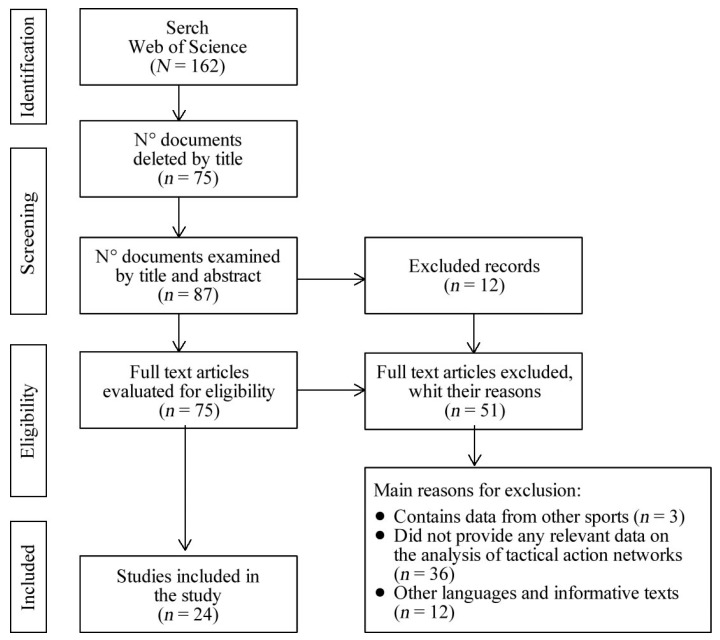
Flow chart of the procedure used in the systematic review.

**Figure 2 ijerph-17-06649-f002:**
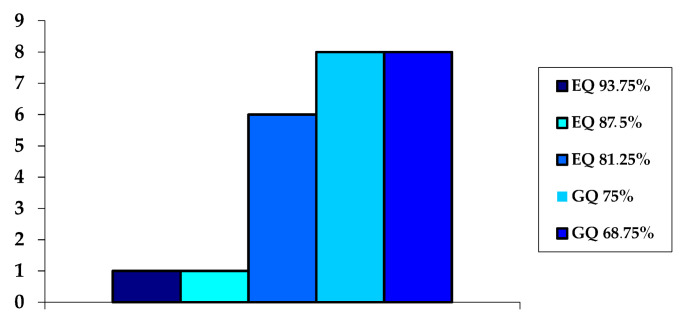
Quality of studies.

**Figure 3 ijerph-17-06649-f003:**
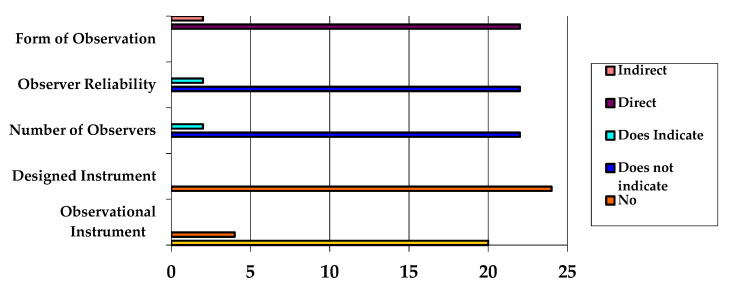
Characteristics of observation methodology.

**Table 1 ijerph-17-06649-t001:** Analysis of the articles selected as a sample of the systematic review.

Study No	Publication Year	Citation	Sample	Independent Variables	Associated Variable	Data Collection System	Key Findings
1	2011	Yamamoto and Yokoyama [22]	Final of the 2006 FIFA World Cup and in an “A” international match in Japan.	Changes in possession, from the passing player to the player receiving the pass.	Time series in the number of triangles by intervals	Unique data set	Greater resilience and survivability, not only in biological networks, but also in communication networks. Few nodes with many connections have been shown to have self-organizing and emergent properties.
2	2012	Grund [23]	283,259 passes between individual players in 760 matches of professional football in the English Premier League, 2006/07 and 2007/08 seasons.	Passes between players to build networks for each team in each game.	Ball possession (in minutes). Network intensity (passing rate). Centralization of the network (compound score)	OPTA Sportsdata unique data set	High levels of interaction lead to higher team performance. Centralized interaction patterns lead to lower team performance.
3	2014	Gama et al. [18]	Six games and 1488 collective attack actions were recorded.	Quantitative and qualitative analysis of attack actions, including: completed passes and crossings involving a total of 4126 intra-team interactions.	Simultaneous movements of each player and interactions between attacking players during the course of the match.	Videotaped match.	Network analysis can be useful to identify specificities in the strategic planning of a team, quantify individual team contributions and interactions through the analysis of relevant actions in football.
4	2015	Clemente et al. [24]	32 national teams for the 2014 FIFA World Cup.	37,864 passes between teammates in 64 football matches enabled the study of the network structure and the performance of football teams.	Number of goals scored and the number of goals received per game	Unique data set	The important contribution of this study lies in differentiating successful and unsuccessful national teams present at the 2014 FIFA World Cup, extending previous studies on network metrics associated with performance variables, and increasing knowledge about the connectivity behavior of teammates in football.
5	2015	Gama et al. [25]	30 matches and 7583 offensive collective actions. 22,518 intra-team interactions in the Portuguese Premier League.	The relevant actions that are executed during the offensive phase.	Level of connectivity between teammates.	Digital video recordings obtained from multiple cameras.	The central players are fundamental in the process of self-organization of the team, since they show a higher level of quality both in the execution and in the reception of passes. Network analysis provides ideas on how creative and organizing individuals might act to orchestrate team strategies.
6	2016	Clemente et al. [26]	Seven matches of the German national football team at the 2014 FIFA World Cup.	Introduction of a software called Performance Analysis Tool for the study of the network structure	The interaction of teammates.	Unique data set.	The network approach allowed the identification of the leading players during the attack process, in the construction (positional attack) and not in the counterattack.
7	2016	Clemente et al. [27]	Thirty-six official matches of the same professional team in the Portuguese Premier Football League.	Analyze goals scored and received by a single team during an entire season using netting methods	The connectivity of the team players and the connectivity of the regions	TV recordings.	The attacking midfield area was identified as the region that contributed most to the goals scored and conceded. Network analysis, as a semi-automatic system, can be a useful and easy-to-use method.
8	2016	Clemente et al. [28]	10 matches of the Spanish League and 10 of the English Premier League.	The top four teams and their opponents in each competitive league were analyzed. A total of 14,738 passes between teammates were recorded and processed.	The players’ tactical positions.	Semi-automatic video tracking system.	The highest levels of centrality were found in outside defenders and midfielders. Central defenders tend to pass the ball to outfielders and midfielders to initiate ball progression
9	2016	Couceiro et al. [29]	The two best teams in the Portuguese Premier League 2010/2011.	999 collective attacking actions were analyzed; the positions of the passes were determined from the players’ positions.	Ball possession. Areas of the field occupied by central players during matches.	Semi-automatic video tracking system.	The net analysis showed the importance of circulation and maintaining possession of the ball by passing to the central player several times.
10	2016	Gama et al. [30]	30 matches, of a Portuguese Premier League team (2010/2011 season).	Interactions resulting from the collective behavior of professional football teams and the influence of ball possession	Successful ball possession and unsuccessful class actions.	Semi-automatic video tracking system.	The analysis of the network showed that professional football teams give particular importance to the movement and maintenance of ball possession by actively collaborating with the central player(s).
11	2017	Gonçalves et al. [13]	The participants included 44 elite male players from age groups under 15 and under 17.	A step-net approach was calculated within the variables derived from the positioning during a simulated match.	Identify the contributions of individual players to the overall outcome of the team’s behavior.	(SPI-ProX, GPSports, Canberra, ACT, Australia) (5 Hz)	This study provided evidence that less reliance on passing for a given player and higher, well-connected passing ratios within the team can optimize performance.
12	2017	Pina et al. [31]	12 matches of the group stage of the UEFA Champions League 2015/2016, Group C.	Investigate whether network density, clustering coefficient and centralization can predict successful or unsuccessful team performance.	The effect of total passes on the success of offensive plays.	Public records of television broadcasts.	A low network density may be associated with a higher overall number of offensive plays, but most are unsuccessful and the high density was associated with fewer and/or more offensive plays.
13	2017	Moreira Praça et al. [32]	18 young football players (age 16.4 ± 0.7), team members participating in national and federated competitions.	The influence of additional players and the playing position on the network properties during small, conditioned 2 x4 min games in football.	Ball possession.	Unique data set.	The condition of the task and the playing position influence the general and individual properties of the network, their analysis allows a better understanding of the characteristics of cooperation during different task conditions.
14	2018	Mclean et al. [33]	108 goals scored at the 2016 European Football Championship in France.	Comparative analysis of passing networks and field locations, as well as the shooting zones involved in the networks.	Measurements of network duration and total connections.	TV recordings.	The state of the games influences the networks from which goals are scored. The current results suggest that measures of centralization from the outside and inside can be used as a measure to determine prominent areas of the field during a match.
15	2018	Arriaza-Ardiles et al. [34]	36 official matches of the Professional Football League (Spain).	Characterize the contribution of players to the team: grouping coefficient and centrality metrics.	Passing/Receiving Chart.	Semi-automatic video tracking system.	Synthesizing the game from the theory of complex graphics and networks from the point of view of nonlinearity allows us to examine the individual role of each player and, at the same time, to understand the performance of the team as a whole.
16	2018	Mendes et al. [10]	132 full official matches.	The passing sequences. The connections between a player and a teammate.	The variation of the overall network properties at different competitive levels.	Semi-automatic video tracking system.	Moderate to strong correlation between general net properties and final score performance variables and goals received. Elite teams had higher overall link values and network density than younger teams. Playing at home significantly increased the homogeneity of teammate relationships during offensive games.
17	2018	McHale and Relton [35]	English Premier League season 2012–13.	Identification of key members of a football team.	Performance of the players of each team throughout the season.	Unique data set. Prozone.	It has been shown that a generalized mixed-additive model can accurately predict the probability of success of a pass in most areas of the field; while finding high levels of volatility in the opponent’s penalty area.
18	2018	Yamamoto and Narizuka [36]	6 matches from the Japanese League (2016 season).	Temporary evolution of the network.	Ball transition.	Unique data set provided by DataStadium Inc, Japan.	The probability of transition in teams with fewer passes tends to have a higher error. Team performance tends to be lower if the weighted ball passing network is highly centralized.
19	2018	Barron et al. [37]	1104 English League matches 2008/09 and 2009/10 seasons	To objectively identify key performance indicators in professional football that influence outfield players using an artificial neural network.	The total actions, the percentage of play, the total goals and assists.	Unique data set. ProZone MatchViewer system and online databases.	It is possible to identify performance indicators through an artificial neural network in players and accurately predict their career path.
20	2019	Buldú et al. [38]	380 matches of the Spanish national league “La Liga” during the 2009/2010 season.	Influence of temporal fluctuations, 50 passes for both teams, paying special attention to goals scored/received.	Connectivity between players.	OPTA single data set.	Increasing the number of passes benefits the overall properties of the passing networks. The dispersion of players around the midfield position of the net is greater when a goal is received.
21	2019	Korte et al. [39]	70 professional matches in categories 1 and 2 of the German Bundesliga season 2017/2018	Identify dominant and intermediary players in football by applying social network game analysis.	Successful and unsuccessful ball possession and the position of the players within the zones of the field.	Semi-automatic multicamera tracking system	Central defenders are identified as dominant players and intermediaries in failed plays. Midfielders are the main intermediaries in successful plays.
22	2019	Kawasaki et al. [40]	9 official matches of Fagiano Okayama - division 2 Japanese League season 2016 and 2017.	Flow items according to the grouping method. Number of passes between different clusters	Successful and unsuccessful ball possession.	Automatic tracking system. Recorded by two high-density cameras	The location of the nodes was determined by grouping the positions of a passer and a receiver with respect to the successful passes. The total passing network metrics indicated the relative level of the number of successful passes.
23	2019	Praça et al. [41]	18 U-17 players from the national class team (Brazil)	Compare tactical behavior, percentage of successful tactical principles and network properties between the highest and lowest aerobic power in young soccer players	Player interactions.	Tactical Evaluation System in Football.	Aerobic power has a limited impact on players’ tactical behavior and network properties, indicating that player’s actions within a small-sized games are mostly limited by other parameters.
24	2019	Diquigiovanni, and Scarpa [17]	380 matches of the Italian “Serie A TIM” 2015–2016 season	Check the effect of playing styles on the number of goals scored.	Space coordinates of specific plays. Connections between nodes.	Unique data set provided by InStat.	15 major tactics were detected. The Dixon and Coles model does not allow the prediction of the final result of a game, the styles of play are available only at the end of the match. The construction of the offensive maneuver from the side of the field has a positive effect on the number of goals scored by a team.

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
