# Peer review of "Passing Networks and Tactical Action in Football: A Systematic Review"

_ijerph, 2020, doi:10.3390/ijerph17186649_

Round 1
Reviewer 1 Report
I have reviewed the article titled “Passing Networks and Aspects Associated with Tactical Action in Football: A Systematic Review”. I consider that the paper is suitable for publishing mainly because it provides valid knowledge for the scientific community, as well as it meets the necessary methodological requirements. However, I suggest some modifications that can make the final document better.
Regarding the introduction, I consider that it is not necessary to modify any aspect.
Concerning the Materials and Method section, I consider:
1. It is highly recommended to introduce the PICOS strategy when formulating the research question well
2. Why is the bibliographic review carried out only until September 2019? It is possible that, in all this time, especially after the confinement in which the number of publications has grown exponentially, something has been published about it and, therefore, it would be advisable to carry out a final review.
3. Concerning the inclusion criteria, it is established that articles with soccer players will be analyzed (criterion b). However, I consider this to be one of the main limitations of the study (and I will return to this later). I think it is necessary to establish some differences depending on the competitive level or skill of the players, or even the gender criteria because otherwise, it would lead to an incorrect interpretation of the results. If this division or limitation were established here, in this section, the number of articles would probably be reduced, but I think the results would be more homogeneous and probably better interpreted.
4. In the data extraction section (page 3, line 92-99) it is not reflected where said table appears. I think it refers to Appendix A, and I think it should be reflected.
About the Results section:
1. I consider it mandatory to modify the flow chart and adapt it to the PRISMA format. http://prisma-statement.org/PRISMAStatement/FlowDiagram
2. The data reflected in the sections “Quality of studies” and “Basic characteristics of the studies included in the review” I consider that it would be convenient to also present them graphically, in such a way as to make the document more pedagogical, at the same time as these results are more clearly reflected.
3. In general, the tables presented in the document, and specifically in this case, Table 1, should be considerably improved in their format. Some suggestions are that being such a large table, the heading should appear on every new page, otherwise, the reader would be lost in understanding.
On the other hand, the order of exposure of the articles is unknown and it is difficult to infer. As the years do not appear, it is not understood why one article appears first and then another. It would be advisable to include the years of publication, as well as to expose the articles following some logical criteria such as the year of publication. This could give a better vision and understanding to readers, observing the evolution of the results, etc.
Lines (articles) should not appear cut in half, with part of the information appearing on one sheet and part on the next. It is necessary to make an effort to layout the table well.
I suggest trying to reduce the length of the table. To do this, I think you could try to remove the "Quality of studies" column (I think it would be more advisable to put this information in the "Quality of studies" section) and reduce the information in the column "Key findings" somewhat improving the written expression. In this way, the table can be better.
4. Finally, in line with the aforementioned on the need to differentiate the studies according to the level of the footballers, in the case of establishing said differentiation, either the number of articles would have been reduced or it would be convenient to expose two different tables depending on the level of the footballers.
5. On page 10 (lines 177 to 182) the most relevant results obtained from the bibliographic review are established as a summary. I consider that this summary is very general and superficial, especially the first two results that are presented. I think you can go deeper and get to a more pragmatic or practical level, in the style of the last two results. For this, I suggest thinking more about giving practical information, useful for coaches. What can a reader expect who approaches this article? This question should lead to thinking about what kind of results can be expected and would be desirable.
6. On page 10 (lines 183-186), again, I am posed with the same question that has arisen earlier. The fact of putting all the results in the same drawer, without putting as filter variables of the level of the players' style, or male and female football, I think leads to a generalization that causes loss of information. I strongly recommend establishing these levels of differentiation in the literature to extract more concrete and precise information.
Concerning the discussion section, the reflections that I add are general reflections, related to the philosophy and generality of the discussion and not specific to a specific point.
1. As a general reflection of the discussion, I think it would be good to extract useful ideas for the coach. What ideas can be extracted from this bibliographic review that helps the coach in the design of tasks, in directing matches? For me, the discussion remains very superficial and impractical.
2. I believe that the research carried out in this area has an important limitation. Probably, I have not captured well the different research designs carried out in it (and therefore, if so, it would be necessary to also analyze this section in the results), but as far as I understand, the different studies carried out analyze the network of passes based on specific matches. I think this can lead to misinterpreting the data and drawing conclusions, specific to a game, and not to a team. For this reason, I understand that it would be very convenient, in the discussion, to establish a discussion about the different designs used and suggest analyzing the network of passes throughout a season, having a deeper knowledge throughout the year, and even, understanding how this network of passes has to be modified depending on the rival, their defense, etc.
3. Another limitation of the studies carried out (not of the bibliographic review) that could be considered, and therefore debated throughout the discussion, is the fact that the research carried out so far, remains on a plane very descriptive, it is researched through sport, but there is no direct transfer for performance improvement or learning, it is not researched for sport.
4. In the “Performance Context” section (page 10, lines 204-205), it is established that the studies are homogeneous concerning the need to study particular movements… Understanding what is being argued, I think this expression should be qualified, because (returning to my argument) I understand that the studies are heterogeneous if the competitive level or the difference between gender is not addressed. There is such a diversity of shows and competitions that, at least to me, all the conclusions and results obtained suggest many doubts. I think this aspect should be debated in the discussion.
5. Again, I think that accompanying the text with images or graphics would be highly recommended (page 10, lines 197-202).
Finally, I consider it mandatory to modify the table in Appendix A. In the current format it is incomprehensible and unaffordable. I suggest pivoting rows and columns and reducing/removing non-relevant information (eg number of authors). Perhaps this way the understanding of said table is better. I suggest adding the suggestions made for Table 1, on headings; as well as establishing a logical order of presentation of the articles.
Author Response
Dear reviewer
We appreciate your comments and recommendations to improve the final version of this manuscript. Below, we provide answers to each of them.
- It is highly recommended to introduce the PICOS strategy when formulating the research question well
We appreciate this suggestion; however, we consider that the PICOS strategy is more adapted to clinical and general health science research.
- Ramirez-Campillo, R., Sanchez-Sanchez, J., Romero-Moraleda, B., Yanci, J., García-Hermoso, A., & Manuel Clemente, F. (2020). Effects of plyometric jump training in female soccer player’s vertical jump height: A systematic review with meta-analysis. Journal of Sports Sciences, 1-13.
- Goes, F. R., Meerhoff, L. A., Bueno, M. J. O., Rodrigues, D. M., Moura, F. A., Brink, M. S., ... & Lemmink, K. A. P. M. (2020). Unlocking the potential of big data to support tactical performance analysis in professional soccer: A systematic review. European Journal of Sport Science, 1-16.
- Clemente, F. M., Afonso, J., Castillo, D., Los Arcos, A., Silva, A. F., & Sarmento, H. (2020). The effects of small-sided soccer games on tactical behavior and collective dynamics: A systematic review. Chaos, Solitons & Fractals, 134, 109710.
- Why is the bibliographic review carried out only until September 2019? It is possible that, in all this time, especially after the confinement in which the number of publications has grown exponentially, something has been published about it and, therefore, it would be advisable to carry out a final review.
In fact, during the course of 2020, some research has been developed in relation to passing networks and tactical actions in soccer. However, we consider that the results obtained up to September 2019 represent an important and significant contribution to knowledge; likewise, we are currently developing the second phase of this systematic review, in which the studies published in 2020 are addressed from a perspective that helps overcome the limitations identified in this review.
- Concerning the inclusion criteria, it is established that articles with soccer players will be analyzed (criterion b). However, I consider this to be one of the main limitations of the study (and I will return to this later). I think it is necessary to establish some differences depending on the competitive level or skill of the players, or even the gender criteria because otherwise, it would lead to an incorrect interpretation of the results. If this division or limitation were established here, in this section, the number of articles would probably be reduced, but I think the results would be more homogeneous and probably better interpreted.
The present systematic review looked only at research whose sample is related to male high-performance soccer players and children's federated sports, this is explicitly stated on page 2 line 74.
- In the data extraction section (page 3, line 92-99) it is not reflected where said table appears. I think it refers to Appendix A, and I think it should be reflected.
In accordance with this assessment, the modification was made and the mention of Appendix A is now on page 3, line 96.
About the Results section:
- I consider it mandatory to modify the flow chart and adapt it to the PRISMA format. http://prisma-statement.org/PRISMAStatement/FlowDiagram
The flow chart was adapted according to the PRISMA format (page 3, line 116).
- The data reflected in the sections “Quality of studies” and “Basic characteristics of the studies included in the review” I consider that it would be convenient to also present them graphically, in such a way as to make the document more pedagogical, at the same time as these results are more clearly reflected.
Taking into account this comment, figures 2 and 3 were constructed, which graphically represent the quality of the studies and the characteristics of the observation methodology of the research works analyzed (page 4, lines 124 and 140).
- In general, the tables presented in the document, and specifically in this case, Table 1, should be considerably improved in their format. Some suggestions are that being such a large table, the heading should appear on every new page, otherwise, the reader would be lost in understanding.
Based on this recommendation, Table 1 was adjusted and the header can be seen at each top of the page (pages 6 to 10)
On the other hand, the order of exposure of the articles is unknown and it is difficult to infer. As the years do not appear, it is not understood why one article appears first and then another. It would be advisable to include the years of publication, as well as to expose the articles following some logical criteria such as the year of publication. This could give a better vision and understanding to readers, observing the evolution of the results, etc.
In this sense, Table 1 was linked to the years of publication of each article and was organized in chronological order. Likewise, on page 5, line 153, the order in which the studies are presented is specified.
Lines (articles) should not appear cut in half, with part of the information appearing on one sheet and part on the next. It is necessary to make an effort to layout the table well.
This consideration was adjusted and the table was organized in such a way that the lines and their contents are not cut in half (pages 6 to 10)
I suggest trying to reduce the length of the table. To do this, I think you could try to remove the "Quality of studies" column (I think it would be more advisable to put this information in the "Quality of studies" section) and reduce the information in the column "Key findings" somewhat improving the written expression. In this way, the table can be better.
Table 1 was modified. As suggested, the quality of studies column was eliminated and this information was included on page 4, lines 119 to 123, and represented graphically in Figure 2 (pages 6 to 10)
- Finally, in line with the aforementioned on the need to differentiate the studies according to the level of the footballers, in the case of establishing said differentiation, either the number of articles would have been reduced or it would be convenient to expose two different tables depending on the level of the footballers.
In view of the need to differentiate the studies according to the level of the soccer players, specific information was included in the section on results, page 11, lines 190 to 195, with regard to children's federated sport and high performance sport.
- On page 10 (lines 177 to 182) the most relevant results obtained from the bibliographic review are established as a summary. I consider that this summary is very general and superficial, especially the first two results that are presented. I think you can go deeper and get to a more pragmatic or practical level, in the style of the last two results. For this, I suggest thinking more about giving practical information, useful for coaches. What can a reader expect who approaches this article? This question should lead to thinking about what kind of results can be expected and would be desirable.
Based on this suggestion, the first two results were adjusted according to their usefulness for the trainers (page 11, lines 180 to 182)
- On page 10 (lines 183-186), again, I am posed with the same question that has arisen earlier. The fact of putting all the results in the same drawer, without putting as filter variables of the level of the players' style, or male and female football, I think leads to a generalization that causes loss of information. I strongly recommend establishing these levels of differentiation in the literature to extract more concrete and precise information.
In this sense and in view of the need to differentiate the studies according to the level of the soccer players, specific information was included in the section of results, page 11, lines 190 to 195, regarding high performance and federated sport for children.
Concerning the discussion section, the reflections that I add are general reflections, related to the philosophy and generality of the discussion and not specific to a specific point.
- As a general reflection of the discussion, I think it would be good to extract useful ideas for the coach. What ideas can be extracted from this bibliographic review that helps the coach in the design of tasks, in directing matches? For me, the discussion remains very superficial and impractical.
Based on this comment, the discussion section was revised and lines 267 to 272 and 283 to 287 were linked on page 13.
- I believe that the research carried out in this area has an important limitation. Probably, I have not captured well the different research designs carried out in it (and therefore, if so, it would be necessary to also analyze this section in the results), but as far as I understand, the different studies carried out analyze the network of passes based on specific matches. I think this can lead to misinterpreting the data and drawing conclusions, specific to a game, and not to a team. For this reason, I understand that it would be very convenient, in the discussion, to establish a discussion about the different designs used and suggest analyzing the network of passes throughout a season, having a deeper knowledge throughout the year, and even, understanding how this network of passes has to be modified depending on the rival, their defense, etc.
To address this suggestion, lines 302 to 304 in the limitations section on page 13 were linked.
- Another limitation of the studies carried out (not of the bibliographic review) that could be considered, and therefore debated throughout the discussion, is the fact that the research carried out so far, remains on a plane very descriptive, it is researched through sport, but there is no direct transfer for performance improvement or learning, it is not researched for sport.
This recommendation was included on lines 305 to 308 on page 13 of the limitations section.
- In the “Performance Context” section (page 10, lines 204-205), it is established that the studies are homogeneous concerning the need to study particular movements… Understanding what is being argued, I think this expression should be qualified, because (returning to my argument) I understand that the studies are heterogeneous if the competitive level or the difference between gender is not addressed. There is such a diversity of shows and competitions that, at least to me, all the conclusions and results obtained suggest many doubts. I think this aspect should be debated in the discussion.
Regarding this suggestion, line 74 of the review (inclusion criteria) specifies that the studies analyzed are those that include a sample of male soccer players, also, it is specified that the level of competition of the sample is high performance and federated sport for children.
- Again, I think that accompanying the text with images or graphics would be highly recommended (page 10, lines 197-202).
The above lines are linked to give a clearer and more detailed explanation of the networks and nodes.
Finally, I consider it mandatory to modify the table in Appendix A. In the current format it is incomprehensible and unaffordable. I suggest pivoting rows and columns and reducing/removing non-relevant information (eg number of authors). Perhaps this way the understanding of said table is better. I suggest adding the suggestions made for Table 1, on headings; as well as establishing a logical order of presentation of the articles.
Based on the assessments and suggestions, Appendix A was modified in its entirety and the suggestions made were added to Table 1 (page 15 and 16).
"Please see the attachment."

Reviewer 2 Report
The systematic review is very good and well designed. I just have a suggestion for the conclusion: be more accurate. A systematic review that does not present a more direct position loses in quality. I don't like this type of writing "These results highlight the need to study the interaction between players, organizing information from the teams within the field, which helps anticipate the competition and generate more options to win, complementing the vision of the game and avoiding loss of information. "at the conclusion of a systematic review abstract.
The conclusion presented at the end of the paper is much better!
The title is relevant, but is it not possible to present one that is more attractive to the reader? Note that in the title "Passing Networks and Aspects Associated with Tactical Action in Football: A Systematic Review" we have the word "aspects associated" and in the question we have the word "factors associated". See that associated factors are different from aspects. What is the intention of the authors? Try to standardize and give a more striking title.Author Response
Dear reviewer
We appreciate your comments and recommendations to improve the final version of this manuscript. Below, we provide answers to each of them.
The conclusion presented at the end of the paper is much better!
Based on this suggestion, the conclusion presented in the summary of the manuscript was replaced by the conclusion presented at the end of the document.
The title is relevant, but is it not possible to present one that is more attractive to the reader? Note that in the title "Passing Networks and Aspects Associated with Tactical Action in Football: A Systematic Review" we have the word "aspects associated" and in the question we have the word "factors associated". See that associated factors are different from aspects. What is the intention of the authors? Try to standardize and give a more striking title.
Following this suggestion, the title a is modified: Passing Networks and Tactical Action in Football: A Systematic Review (page 1, lines 2 and 3).
"Please see the attachment."

Round 2
Reviewer 1 Report
I believe that this article, in its current format, can be published, since it meets all the necessary conditions for it.
I have only observed a small error, in the layout of the table, figure nº 1, that cannot be seen in its entirety.